# Transgenerational Epigenetic Inheritance of Traumatic Experience in Mammals

**DOI:** 10.3390/genes14010120

**Published:** 2023-01-01

**Authors:** Jana Švorcová

**Affiliations:** Department of Philosophy and History of Science, Faculty of Science, Charles University, 128 00 Prague, Czech Republic; jana.svorcova@natur.cuni.cz

**Keywords:** transgenerational epigenetic inheritance, stress, trauma, DNA methylation, RNA, HPA axis

## Abstract

In recent years, we have seen an increasing amount of evidence pointing to the existence of a non-genetic heredity of the effects of events such as separation from parents, threat to life, or other traumatising experiences such as famine. This heredity is often mediated by epigenetic regulations of gene expression and may be transferred even across several generations. In this review, we focus on studies which involve transgenerational epigenetic inheritance (TEI), with a short detour to intergenerational studies focused on the inheritance of trauma or stressful experiences. The reviewed studies show a plethora of universal changes which stress exposure initiates on multiple levels of organisation ranging from hormonal production and the hypothalamic-pituitary-adrenal (HPA) axis modulation all the way to cognition, behaviour, or propensity to certain psychiatric or metabolic disorders. This review will also provide an overview of relevant methodology and difficulties linked to implementation of epigenetic studies. A better understanding of these processes may help us elucidate the evolutionary pathways which are at work in the course of emergence of the diseases and disorders associated with exposure to trauma, either direct or in a previous generation.

## 1. Introduction

Since 1925, when DNA methylation was discovered in bacteria [1], epigenetics had come a long way and became a well-established field of science. The term ‘epigenetic’ was defined in several different ways since it was first coined by Waddington [2]. In the initial sense, it was essentially bound to Waddington’s model of epigenetic landscape (Figure 1), which depicts the space of possible states which a cell, tissue, organ, or organism—here represented by a marble—can possibly find itself in. The marble moves through a landscape which canalises the development into specific pathways and makes it resistant to perturbations. In this model, epigenotype is the sum of interaction both among genes themselves and among the genes and the environment. The term ‘epigenetic’ has also been used in some evo-devo studies to describe certain plastic evolutionary processes which are not yet genetically ‘hardwired’. Such epigenetic processes include chemical, electrical, or mechanical interactions of primitive cells with the environment and primitive cells among themselves, but also interactions of primitive cell metabolisms with the external environment [3].

The prefix epi-means ‘beyond genes’ and indeed, generally speaking, epigenetics studies all non-genetic evolutionary variation, that is, everything that is not part of the DNA script. Epigenetic inheritance can thus refer to the inheritance of entire cellular structures (organelles, such as centrioles, mitochondria, plasma membranes, cilia etc.), but also to the inheritance of for instance microbial symbionts. In a narrow sense, (molecular) epigenetic inheritance is nowadays usually taken to refer to the inheritance of chromatin structures and modifications (both DNA methylation and several histone modifications) as well as to various types of RNA and exosomes. This is the definition we will work with. (For details on these processes, see Box 1).

Epigenetic inheritance across several generations has been known under various names, including dauermodification, paramutation, or lingering mutation [4,5,6]. In plants, it has been known for over a century. Modern studies, for instance research on *Arabidopsis thaliana* [7], *Taxaracum officinale* [8], or *Linaria vulgaris* [9]—which showed the inheritance of DNA methylation—further contributed to this direction of research. Along similar lines, epigenetic inheritance has been demonstrated in yeast *Schizosaccharomyces pombe* (inheritance of histone modification) [10], *Drosophila melanogaster* (trans-inheritance of the chromatin state) [11,12], or in several studies on the nematode *Caenorhabditis elegans* ([13], inheritance of RNA molecules [14], and histone modification [15,16]). 

There are also known examples of transgenerational epigenetic inheritance (TEI) in mammals, including the almost notorious example of intracisternal A-particle and metastable epiallele *A^vy^* in mice [17], which leads to diverse coat colouring in genetically identical individuals. This intracisternal A-particle (see also Box 1) can be modified by methylation, which depends on environmental signals (maternal diet with sufficient methyl donor such as folic acid or B 12) and results in the brown pseudoagouti phenotype [18,19]. One can observe, meanwhile, that coat colour is transgenerationally inherited through the maternal germline [20] and the agouti phenotype is associated with various diseases [21]. The *Axin^Fu^* epiallele is a similar case [22,23]. (For review on metastable epialleles, see [24].) In a related example, the exposure of guinea pigs to heat led to a transgenerational epigenetic effect in the male progeny, where the DNA methylation pattern changed in relation to the expression of thermoregulating genes in the liver [25]. In a similar way, heat conditioning of chicks led to a transgenerational heat and immunological resilience, which was once again accompanied by changed methylation patterns [26].

As indicated above, nutrition is another subject of interest to TEI studies. The *A^vy^* studies were followed by numerous further studies on this subject. In pregnant rats, a protein restriction diet led to a 20% lower methylation of hepatic peroxisome proliferator-activated receptor α gene (*PPARα*), which increased the activity of this gene [27]. It has also been described that high-fat diet can lead to increased female body size in the paternal lineage in mice [28], while folate and methyl-donor diet restrictions can alter the DNA methylation patterns in rats [29]. 

Exposure to drugs also has an impact on TEI. One of the first studies in this area had shown that parental smoking (before child’s age of 11 years) was associated with a greater body mass index at 9 years in sons, but not daughters [30]. Similarly, the exposure of adolescent rats to cannabinoids had a transgenerational effect observable in morphine sensitisation [31], while transgenerational effects after foetal alcohol exposure persisted in the male (but not female) line until the F3 generation [32]. The observed effects involved increased gene methylation of the *POMC* gene (proopiomelanocortin gene which is active in the stress-regulation axis) and its reduced expression, increased corticosterone response, and elevated anxiety in behavioural testing.

Another important chapter of environmental epigenetics is the exposure to toxins, for example to vincozolin, which influences the development of testis and can later lead to premature apoptosis of sperm cells in adults [33]. Pregnant mice exposed to vincozolin during the period of foetal gonadal sex determination gave birth to infertile sons with various organ defects [34,35]. This effect persisted through DNA methylation pattern changes into generation F4 (subsequent generations were born from IVF). It also had an impact on sexual selection, where naive, not exposed, control females did not want to mate with defective males [36] even when the male exposed to vincozolin in utero was already four generations away (the reverse did not hold, though—males were not so selective). Later studies associated these effects in generations F3 and F4 with changes to the types of small noncoding RNAs in sperm [37]. Another study, though, found that epigenetic changes after exposure to vincozolin were erased in subsequent generations [38].

Similar effects have been documented in relation to other toxicants. For instance, when pregnant female rats were exposed to bisphenol A, the males in F3 had defective testes [39]. After exposure of mice to dioxin, another study [40] showed a transgenerational effect of reduced fertility and increased incidence of premature birth. Naturally, mixtures of toxicants can likewise have a transgenerational effect, as in the case of ovarian disease in rats, which is similar to human polycystic ovary syndrome [41].

As noted above, TEI has been described in various mammalian models. This review focuses on TEI of traumatic experience with emphasis on mammals, chiefly rodents and humans, because in them, one can really talk about traumatic or stressful experience or its analogues. For the purpose of the present overview, we define trauma as a distressing experience or adverse event that can lead to emotional and/or behavioural responses. Stress is a physiological response to a stressful stimulus which involves the hypothalamic-pituitary-adrenal (HPA) axis. This axis plays a significant role in many processes related to environmental cues involving digestion, energy storage, immunity, and emotional responsivity. Its dysregulation is associated with elevated cortisol levels and consequently also with changes in neurogenesis, neural density, and both glio- and synaptogenesis, possibly leading to changes in cognition and behaviour or to psychopathological or affective disorders [42]. In the following, we examine studies involving exposure to stress and trauma of parental generations and subsequent intergenerational and transgenerational inheritance of their impact.

Box 1Possible epigenetic processes.Epigenetic modifications can be induced randomly but also environmentally, and they can be inherited over a number of generations. A vast number of macromolecules (nucleic acids, proteins, sugars, and possibly all proteins in the cell, [6]) or even entire structures (chromatin, as mentioned above, but also cytoskeleton, glycocalyx, intercellular mass), are epigenetically modified in the cell. This box focuses on three basic epigenetic processes: DNA methylation, histone modification, and categorisation of various RNA molecules (including, e.g., the role of exosomes). These processes are responsible for the regulation of genome transcription, that is, for cellular differentiation and the development of organism in general. They can be inherited both mitotically and meiotically. Epigenetics is thus actually the study of Aristotelian epigenesis in the Waddingtonian sense, i.e., the study of emergence of a body form from a single zygote, which involves various settings of DNA transcription through differentiation of different cell types. For more detailed information on the specific mechanisms of TEI in various model organisms and their signalling interactions, see [43].
**DNA methylation**
DNA, as well as different types of RNA, can undergo various reversible modifications. One of the earliest epigenetic modifications discovered was DNA methylation, where a methyl group is attached by methyltransferase to cytosines, giving rise to 5-methyl cytosine (5 mC). Adenines can be also methylated (in bacteria, fungi, *Drosophila* or mammals [44]). In addition, 5 mC can be also oxidized by Tet proteins to 5-hydroxymethyl cytosine, 5-formyl cytosine, or 5-carboxy cytosine [45]. As a rule, though, methylation of cytosines (in combination with the corresponding modifications of histones) leads to the silencing of a given DNA region via binding of various proteins to the promotor of the gene. Frequently, whole stretches of DNA (CpG islands) are modified in this way. Methylation can take place *de novo* (in case of vertebrates by DNMT3) or be maintained across several cell divisions by replication of the modified parental strand. In vertebrates, DNMT1 can recognise DNA strands with unmodified cytosines and methylate them.In mammals, DNA methylation is also associated with genomic imprinting, which involves a differential expression of paternal and maternal alleles of the same genes. The genes concerned are often related to foetal growth, where maternal alleles balance the products of expression of paternal alleles (e.g., insulin-like growth factor 2 receptor gene [46]). Paternal alleles tend to promote foetal growth, which is unfavourable for the maternal organism. Therefore, a balanced expression depending of imprinted genes from both sexes is of pivotal significance, because uniparental diploid embryos stop developing at the implantation stage [47]. Disrupted genomic imprinting in humans leads to various disorders involving hormonal dysregulation, aberrant pre- and postnatal growth, hypotonia, abnormal behaviour or mental retardation, psychiatric disorders, sleep disorders, and many other symptoms [48]. Regulation of genomic imprinting also involves RNA molecules and histone modifications (e.g., H3K27me3, i.e., the trimethylation of histone 3 on lysine 27 [47]).In the course of their development, mammals undergo two main reprogramming events: one during the early stages of embryonic development, in preimplantation embryos, and the other in the primordial germ cells (PGCs) [49]. DNA methylation marks are deleted and later newly rewritten. Such reprogramming is viewed as adaptive: its function is to remove epigenetic signatures acquired randomly or from various environmental cues so that the development is not disrupted. Moreover, reprogramming leads to cellular totipotency [50] required for development. Some regions, such as imprinted genes, intracisternal A-particle elements (IAPs), or LINE 1 elements, can apparently avoid such deletion in the preimplantation period [51]. During reprogramming in PGCs, the DNA methylation is erased again (including imprinted genes which allows a re-establishment of sex specific genomic imprints). However, a study reports that a significant fraction of IAP (but not LINE 1) remains methylated in PGCs [52,53]. This fact can explain the inheritance of metastable epialleles such as *A^vy^* in which an insertion of variably methylated IAP retrotransposon leads to ectopic expression of the agouti gene. Sequences in proximity to IAPs have consistently high methylation levels throughout of all developmental stages of PGCs [54]. Other research had concluded that IAP are remethylated after fertilisation [24]. Genome-wide DNA methylation studies show that over 40% of 5 mC in PGCs escape demethylation [53]. These regions are usually the most active and mobile, and therefore possibly highly mutagenic, repeat sequences of a type corresponding to IAP. Seisenberger et al. [54] reported that a significant proportion of CpG islands remains methylated during various developmental stages of PGCs (more in the case of sperm), representing a possible carrier of TEI.A study by Kremsky and Corces suggests [55] that some regions of the CpG sites bound by transcription factors remain methylated during the reprogramming stages. In mice, there are several transcription factors which bind histone modifications (e.g., H3K9me2) and block TET methylcytosine dioxygenase 3 activity (which converts 5-methylcytosines to 5-hydroxymethylcytosines), thus maintaining DNA methylation marks [56]. This indicates that transcription factors can also act as (trans) mediators of epigenetic inheritance.
**Histone modification**
The DNA is wrapped around a heterooctamer of histones H2A, H2B, H3, and H4 (each in two copies). The whole octamer is stabilised by histone 1 protein. Histone terminal tails can be specifically modified by various enzymes, especially amino acids lysine, threonine, tyrosine, and serine. Possible modifications include phosphorylation, acetylation, glycosylation, mono, di-, and tri-methylation, ubiquitination, and citrullination [57,58]. Modifications can also be erased, rewritten (histone modifiers can be viewed as writers, readers, or erasers [59]), or become the substrate for a specific enzyme action, usually condensation or de-condensation of DNA from histones in order to access DNA for transcription or repair. Histone modifications are conservative among the taxa. For instance, methylation of lysine 4 on histone 3 (H3K4), H3K36, or H3K79 is usually associated with active chromatin [57]. However, H3K9me3, H3K27me3, and H4K20 are usually involved in chromatin silencing and associated with DNA methylation [57]. It is, moreover, known that H3K27me3 [60] or siRNA-induced H3K9me3 [10] can be inherited epigenetically through several generations. In sperms, histones are usually replaced by protamines, but recent evidence shows that around 10% of histones in humans and 1% in mouse sperm do in fact remain [61]. Protamines can likewise be modified [62]. Histones are multiplied and specifically modified, and each (human) cell thus contains around 30 million of uniquely configured nucleosomes.
**RNAs**
RNA molecules are generally divided into long noncoding RNAs (around 200 nucleotides), which form a kind of scaffolding for various proteins. These complexes also activate or silence gene expression. Another category consists of the small noncoding RNAs of many different types, which have many different functions. For example, microRNAs (around 22 nucleotides) silence the RNAs in the cytoplasm, while piwi RNAs (in animals) or small interfering RNA (siRNA, in *S. pombe*, *C. elegans*, or *D. melanogaster*) inactivate certain stretches of DNA, mostly unpaired strands or retrotransposons in the sex cells (by methylation of cytosines) but can also silence RNAs transcribed from these retrotransposons. Small interfering RNAs in plants are known to form the paternal epigenetic signal leading to DNA methylation and gene silencing [63]. Transfer RNAs (tRNAs) are known for their role in translating RNA into proteins. Their fragments, tsRNAs (around 30 nucleotides), are dominant in mammalian sperm [64,65] and it is known their amount in sperm can be influenced by diet [66]. They may also be responsible for inheritance of the specific phenotype of mouse tail [67] or for intergenerational inheritance certain metabolic disorders [68,69,70,71]. For detailed information on the role of sperm RNA see [72].All of the abovementioned regulatory RNAs can also be epigenetically modified. For instance, RNA adenosines can be methylated to form 6-methyladenosine. The newly found variety of RNA types in sperm and its modifications inspired the proposal of the ‘sperm RNA code’ hypothesis [71].
**Exosomes**
Small noncoding RNAs are often transported around the body by exosomes, which are vesicles often generated by the somatic tissue. For example, sperm cells of mice, as they mature in the epididymis, fundamentally change the content of their RNA molecules as they progressively acquire these RNA-loaded exosomes during their travel through the epididymis. The somatic tissue thus communicates with germ cells and can transmit information about the environment which the organism has encountered [73,74] (as it happens, these processes are strikingly reminiscent of Darwin’s theory of pangenesis). These RNAs then have complex and diverse functions during early development, where—as noted above—they regulate other epigenetic processes, such as DNA methylation or modification of histone and other proteins. It has been proposed that RNA molecules are the most likely mediator of TEI [75]. The discussion regarding whether the DNA methylation is a true signal of TEI when it is erased and rewritten during epigenetic reprogramming thus need not be relevant anymore, because RNA as a heritable molecule can be sufficient for catalysis of other epigenetic processes, such as chromatin modification. So far, this is just a speculation because, as Bohacek and Rassoulzadegan [65] point out, it is still not clear whether RNAs are transmitted to the sperm only via epididymosomes or also via circulating factors in blood or in consequence of transcriptional changes in the sperm. Moreover, epididymosomes can also be changed via external environmental changes [76].Possible epigenetic mechanisms also include certain self-assembling structures, such as prions [77], or self-propagating trans signals maintained through self-sustaining feedback loops [78], such as *lac* operon bistable system in *E.coli* [79].

## 2. The Methodology of TEI Studies 

Because transgenerational epigenetic inheritance in mammals also involves genetic, ecological, and in the case of humans even cultural factors, scientists wishing to study TEI usually need to take a series of steps.

### 2.1. Mutation

First of all, one needs to establish that the observed phenotypic trait one wishes to study is not influenced by a change in DNA information. If it is, the studied phenomenon would not be a case of epigenetic inheritance. Importantly, if the genes underlying the desired phenotypic trait are unknown, one cannot rule out DNA mutation. This particular challenge can be met for instance by using inbred mice as a model.

### 2.2. Social Transmission

One must also exclude social transmission. A stressed individual can influence naïve individuals by its own changed behaviour and these changes can be fast. On the other hand, isolation itself can be a stressful factor [80]. Researchers who want to eliminate the possibility of social transmission tend to resort to cross-fostering, which is in some studies done in conjunction with in vitro fertilisation (IVF), embryonic transfer, or insemination.

### 2.3. Differentiate between Prenatal (Maternal) Influences, and Intergenerational and Transgenerational Inheritance

Mammalian females communicate with their offspring during pregnancy through the placenta and pregnancy lasts for a relatively long time (19–21 days in mouse, 21–24 days in rats, 9 months in humans). The mother can thus easily influence her offspring already in utero either via changes in uterine artery blood flow or though dysregulation of the maternal-foetal HPA axis via (sometimes excessive amounts of) maternal glucocorticoids. Catecholamines, mineralocorticoids, and sex steroids are also possibly involved in this process. In humans, foetal HPA axis is fully developed by 22nd week of gestation [42]. The epigenetic modifications which are the main focus of this paper are thus key mediators and stabilisers of the prenatal effect. In summary, one must carefully distinguish between prenatal, postnatal, or intergenerational influences on the one hand, and genuine TEI on the other.

The prenatal effect is a process where the offspring are affected by the environment through the mother in utero, but the epigenetic effect is not transmitted to subsequent generations. Logically, this effect involves a different number of generations for females and males. Imagine a signal from an environment which we know can leave an epigenetic mark. In the case of a pregnant female (Figure 2a), up to three generations can be affected at the same time! The first generation is the mother herself. The offspring she carries in her womb are the second generation. This offspring, however, also have their own sexual and somatic lineage already in place, since it forms relatively early in the development. We must therefore also consider the possibility of the environmental influence affecting the offspring’s germ cells. It is therefore only the fourth generation (F3, the mother being F0), where one can tell whether the effect studied is a genuine case of TEI. For generations F1 and F2, we are only talking about a prenatal effect because we cannot exclude the possibility that the cells were affected by a particular environmental effect all at the same time (Weismann called this phenomenon ‘parallel induction’ [81]) and the effect was then transmitted by the germ cells. 

In males (Figure 2b), one can determine whether TEI is at play already in the third generation (F2; with the father being F0). The signal from the environment may have affected the father (F0) and the sex cells (F1) he carried in the testes at that time (this is, once again, a prenatal influence). If the effect is manifested in his grandchildren (F2), we can then talk about TEI. Since studies with males require fewer generations and lack the intrauterine factor, experiments are lately done mostly with males and their sperm. As a result, much less is known about epigenetic inheritance through oocytes.

Intergenerational epigenetic inheritance overlaps with both the prenatal and postnatal effect: in this case, exposure of the parental generation (F0) affects also the offspring of the next generation (F1).

### 2.4. Type of Epigenetic Process

The next step is to determine which epigenetic process is responsible for transmission of the studied phenotype. If this is possible, one can use this epigenetic factor (possibly RNA molecules) on naïve individuals.

### 2.5. Behavioural Tests

It is known that in mice, stress or a traumatic experience leads to the emergence of a traumatised phenotype with changed social behaviour. Mice can display increased levels of anxious behaviour: they spend less time exploring other mice, show repetitive behaviour, spend a long time cleaning themselves, etc. Anxiety in mice is usually tested with two standardised tests, the elevated plus maze and the forced swim test (Figure 3).

## 3. Traumatic Experience and Epigenetic Mediators: Intergenerational Studies

### 3.1. Stress, the HPA Axis, and Glucocorticoid Receptor (GR)

While this review focuses on TEI, in this section we introduce some intergenerational studies, because they show certain universal processes present in stressed or traumatised phenotypes on several levels of biological organisation. The study of stressful experiences or trauma recognisable on an epigenetic level probably started with papers on the intensity of maternal care and possible epigenetic effects mediating the pathway of corticosteroid response [84,85,86]. Intensity of maternal care is associated with the number of receptors for glucocorticoid hormone in the hippocampus, and these changes are epigenetically mediated. In rats, an enhancer of the *GR* gene is not methylated when pups are born but already one day after birth this locus is being methylated. In pups that receive sufficient grooming, that is, care from their mother where the mother is not only present but also stimulates them by touch, these methylation marks are lost again, and the gene is transcribed more. Such pups also have more GRs in their brains and can better cope with stressful environments than their peers who receive less maternal care [86]. This traumatised phenotype can be further transmitted via maternal behaviour into the next generation. Pups traumatised by poor care have symptoms similar to human PTSD (which is also often associated with the functioning of GR) and score worse on classic rodent experiments than their well-cared-for peers. This also shows that a single locus can be epigenetically realtered during a specific developmental window. 

GR plays a central role in several processes including cardiovascular function, stress-response regulation, immunity, metabolism, reproduction or development [87]. Changes in GR sensitivity, together with changed cortisol levels, are the most studied factors investigated in organisms that had been exposed to trauma or stress. They can have a long-term impact on HPA axis modulation, which starts in utero when glucocorticoids impact brain development and thus also the development of the HPA axis. Other hormones can be also involved. Females and males of dams exposed to chronic stress had decreased basal concentrations of corticosterone (both sexes), while research found elevated juvenile oxytocin and decreased adult prolactin in females [88]. Nevertheless, GR is probably most studied case.

The GR, and especially its decreased amount in hippocampus, is associated with a higher activity of the HPA axis and in humans consequently with psychopathological conditions including suicide, schizophrenia, or mood disorders [89]. A study on the effect of prenatal stress in rats (gestational days 12–18) in three successive generations revealed gradually elevated sensitivity of HPA axis, anxiety-like and aversive behaviour in adult maternal line. The brains of the females had reduced neural density in the prefrontal cortex and hippocampus as well as changed patterns of gene activation in genes influencing synaptic plasticity, maturation of neurons, and arborization [90]. Bohacek et al. [91] reported that the offspring of male mice that experienced postnatal traumatic stress had impaired synaptic plasticity, neuronal signaling, and long-term memory in adulthood. Similarly, post-mortem analyses of hippocampus of suicide victims with a history of abuse revealed hypermethylation of the *GR* locus and its reduced expression, suggesting that similar phenomena may persist into adulthood [92]. Childhood abuse in women is associated with HPA sensitivity [93] and decreased hippocampal volume [94]. Cord blood from mothers who suffered from depression and anxiety during the third trimester of pregnancy revealed an increased methylation of the *GR* gene and, in their children, increased levels of salivary cortisol at the age of three months [95].

It has also been demonstrated that RNA molecules serve as mediators of intergenerational olfactory fear conditioning experiences in mice [96]: RNA acquired from conditioned F0 males that was injected into naive zygotes mediated a transmission of enhanced neuroanatomy and sensitivity to a specific odour. Gapp et al. [97] had shown that inheritance of the traumatised phenotype in mice is mediated by two types of RNA molecules acting in conjunction: long RNAs injected into a naive zygote lead in the developing individuals to a later tendency to overeating, insulin sensitivity, and risk-taking in adulthood, while the short RNAs seem responsible for a tendency to depressive behaviour and greater overweight.

All of the examples mentioned in this section are not TEI studies, but they do point to an involvement of universal phenomena associated also with TEI.

### 3.2. War Experience and Famine

A well-known example of epigenetic inheritance in humans is the Hungry Winter, that is, the Dutch famine in the winter of 1944/1945. A sample from this population had been included in a long-term study on several generations. It showed that people exposed to the famine in utero during the sensitive window of early gestation had a higher occurrence of heart disease [98], higher blood pressure at a younger age [99], and an increased risk of schizophrenia [100]. Moreover, women who were exposed to the famine in utero reported more often that their progeny (F2) were in poor health in later life and had a tendency to obesity [101]. What is not clear is the extent to which these effects are just intergenerational, i.e., not a case of TEI, and some studies indeed report it as a transgenerational effect [101,102]. Along similar lines, data from the Överkalix cohort showed sex-specific effects when the grandfather’s experience of famine was associated with a mortality rate of their grandsons that was more elevated than in the case of grandmothers and granddaughters [30]. Similarly, people who experienced famine during the siege of Leningrad, either as children or in utero, had a higher prevalence of hypertension and shorter telomere length [103].

A review by Lumey et al. [104] showed that there is a consistent relation between famine experienced in utero and adult body size together with health consequences such as diabetes or schizophrenia. In contrast, it had been shown that undernutrition of the grandfathers or grandmothers around the period of slow growth in humans (around 9 years of age) was associated with higher mental health scores in the third generation, that is, in their grandchildren [105]. The timing of exposure to trauma or stress within the life of an individual may also play a significant role in the outcome.

The impact of traumatic experience on people who experienced the Holocaust has been studied by Rachel Yehuda [106]. Her studies focus mostly on intergenerational transmission. They show, for instance, that the offspring of Holocaust survivors have a higher prevalence of PTSD and other psychiatric diagnoses than the controls do, although these descendants did not experience traumatic events themselves [107]. Another study showed a similar effect for grandchildren of Holocaust survivors and the association of this effect with hypomethylation of the gene for cortisol [108]. Maternal (but not paternal) PTSD was also associated with a higher glucocorticoid sensitivity in the offspring of Holocaust survivors [109]. Kertes et al. [110] show the effect of chronic maternal stress and war trauma on methylation of key genes which regulate the HPA axis (*FKBP5, NR3C1, CRHBP* and *CRH*). Elevated cytosine methylations of *FKBP5* in Holocaust survivors and their adult offspring have also been demonstrated by research conducted by Yehuda [111].

In the case of Holocaust, the effects can be just as easily explained by behavioural transmission, which can likewise leave an epigenetic mark. It is highly likely that, in natural systems, the processes of social transmission and epigenetic modifications mutually reinforce and feed into each other. In humans, it is extremely difficult to exclude explanation by social transmission. For instance, in the case of offspring of Holocaust survivors, although they were not directly affected by the same experience as their parents, living with a traumatised individual who had been affected by such traumatising experiences can in itself have an impact. The offspring’s behavioural reaction can be triggered by feelings of guilt or excessive identification in reaction to being exposed to the parents’ (or even the whole community’s) experience from their narratives. Aside from that, the parents may display symptoms of PTSD and the children may mirror them. Children of Holocaust survivors for instance often felt rejected by a parent, felt inappropriately high level of responsibility for a parent at a young age, or guilt over the losses their parents experienced in their lives. Aside from that, parents who experienced the Holocaust may downplay their children’s life experiences and contrast them to the Holocaust or may instil in them an exaggerated fear of the outside world and a general feeling of distrust [112]. It all depends on the manner and level to which the parent was able to cope with the abnormal experience and how that was then reflected in the parent’s behaviour.

Nevertheless, it ought to be noted that a meta-analysis by van IJzendoorn et al. [113] found no signs of secondary traumatization in children of Holocaust survivors: only studies on those who already had a clinical diagnosis related to mental health showed this effect.

Similarly, Gapp et al. [114] showed in case of mice that separation of newborn offspring from their mothers can be beneficial. In fact, males (but not females) whose mothers were stressed and separated from them shortly after birth showed a greater behavioural flexibility and better goal-directed behaviour in adulthood than the controls did. For instance, they learned more quickly to select a preferred food source although they had to wait for it for a longer time, and they quickly learned changed rules of the game. These changes were accompanied by histone modifications of the mineralocorticoid receptor gene (which is in humans associated with depression), and its reduced expression in the hippocampus, as well as heritable changes in DNA methylation in the sperm. The authors believe that early traumatic experience is not necessarily evaluated as negative or positive—it is assessed contextually. In this case, the experience of separation from mother may have promoted an adaptive response in the body which manifested itself in adulthood as an increased ability to cope with adversity.

Similarly, studies from Israel report that subsequent generations of Holocaust survivors demonstrate resilience, do not suffer from a higher incidence of psychopathology [115], and some individuals in fact report greater life satisfaction. A similar pattern has been found specifically among grandchildren of Holocaust survivors [116].

## 4. Transgenerational Epigenetic Inheritance

### 4.1. Fearful Experiences

A study on heritability of olfactory preferences in mice [117] had attracted significant attention. In this study, researchers repeatedly exposed male F0 generation mice to electric shocks in the presence of the smell of acetophenone. Each shock lasted quarter of a second, was 0.4 mA in strength, and the shocks were repeated five times with three-minute breaks at various times for three days. The male mouse subjects became hypersensitive to the smell of acetophenone. Surprisingly, though, their offspring in F1 and F2 also showed hypersensitive reactions to the presence of acetophenone although they had never been exposed to any painful conditioning and other odours did not have a similar effect on them. After obtaining the methylation profile from the sperm of all three generations, the researchers observed a demethylation of the *M71* receptor gene, which is a receptor for acetophenone. Increased activity of this locus due to demethylation thus led to an increased level of these receptors in the brain and an increase in the size of the olfactory bulb for acetophenone, i.e., the area of the brain which processes specifically just this odour. The control group of males that were exposed to acetophenone without the presence of electroshock did not show these changes in the brain or their DNA.

To rule out social transmission, the researchers used cross-fostering experiments (the offspring of traumatised fathers were raised by other parents) and artificial insemination (they used the sperm of conditioned males to inseminate naïve, i.e., control, non-traumatised females at the other end of the campus). These verifications yielded the same result, i.e., offspring hypersensitive to the smell of acetophenone who had—in comparison to controls—an enlarged area of the brain that is responsible for processing the smell. Based on a commonly used cognitive behavioural strategy called extinction [118,119], the authors later showed that the effects of traumatic experience can be reversed by retraining, specifically by positive conditioning in a safe environment. Mice exposed to a particular signal without any negative conditioning gradually became desensitised to the odour. They also lost the epigenetic modifications associated with this experience and the abovementioned physiological changes in the brain likewise disappeared. Similarly, a study that focused on environmental enrichment, i.e., on improved environment in which mice are kept in conjunction with increased physical activity, had shown that these changes had a positive effect on synapse plasticity, learning ability, and memory [120]. This study has also described novel microRNA molecules responsible for transmission of this effect between generations, but in this particular case, we can speak only of an intergenerational effect.

### 4.2. Separation Trauma

Separation trauma and its possible role in TEI has been studied by Isabelle Mansuy and her team [121]. They repeatedly separated male pups at unexpected and irregular intervals from their mothers. When the mothers returned, they were stressed out and consequently ignored their pups. The study showed that the males later exhibited a behaviour pattern analogous to human depressive behaviour and were more anxious in adulthood. Mansuy and her team moreover identified ncRNA molecules in the sperm of the traumatised males which are probably responsible for transmission of the traumatised phenotype. To rule out social transmission, they injected these ncRNAs into the cells of embryos obtained from non-traumatised parents—and individuals that developed from these embryos displayed the same behavioural patterns as the traumatised males. This is the first time it has been shown that behaviour can be inherited through injection of ncRNA. Again, social transmission was eliminated: the traumatised males mated with naive females but were subsequently excluded from the experiment and thus had no contact with the offspring. 

In this study, the researchers also introduced crossover experiments—with analogous results. In conclusion, one can see that in all these procedures the traumatised phenotype always reappeared in subsequent generations. This study had thus demonstrated a genuinely transgenerational transmission which lasted in some cases for up to five generations. In nature, social transmission undoubtedly plays an important role in these processes but in this instance, the researchers only wanted to test for TEI. Similarly, in a mouse model, Rodgers et al. [122] showed transgenerational epigenetic inheritance of parental stressed phenotype was mediated by nine types of microRNAs injected in naïve zygotes. 

Isabelle Mansuy and her team [123] also worked with a group of orphans from Pakistan aged 7–12 years. These children reportedly had higher levels of miRNAs (in particular miR-16 and miR-375) in their blood serum. These RNAs were also found in higher amounts in the sperm of older individuals (18–25 years) who had a similar life experience (loss of a parent). It is thought that these molecules may persist in the body into adulthood. 

Van Steenwyk at al. [124] demonstrated a consistency in TEI of behavioural and metabolic phenotypes up to the third (transmission of depressive behaviour) and fourth (transmission of glucose dysregulation and risk-taking behaviour) generation of paternal postnatal trauma in mice.

## 5. Back to Methodology: Difficulties with TEI Studies

Even if the right methodological procedures are in place (see Methodology of TEI), studies on TEI face numerous challenges. Some critics [125] note that the studies are usually done on very small sample sizes in both humans and rodents. This is linked to the type of model subject; in a similar way, it is difficult to acquire much evidence on TEI in humans, both for ethical reasons and time requirements of such studies (i.e., data up to F2 or F3 generation). Not surprisingly, therefore, studies based on mammalian models thus lack robust replication and, as mentioned above in connection with the vinclozolin exposure or maternal separation, the outcomes of such studies can be quite incompatible. 

As Bohacek [80] emphasises, epigenetics is a highly interdisciplinary field which involves molecular biology, psychology, endocrinology, neurobiology, and several other disciplines. The methodological standards used (sample size, chosen strain) and the techniques, terminology, and language vary greatly. The breeding style of model animals, their nutrition, types of stressors, specific details of the IVF practices or behavioural testing, etc., can have an impact on changes of patterns of epigenetic modifications and consequently on inconsistencies between the studies. As if that were not enough, many studies also show that there are sex-specific differences in epigenetic and behavioural outcomes.

In addition, when one wants to avoid social transmission, the use of IVF in association with the stimulation of ovaries, triggering of superovulation, manipulation of oocytes in vitro, or other procedures can in themselves lead to artificially introduced epigenetic changes [80] because these procedures take place during a highly sensitive period of epigenetic reprogramming.

## 6. Discussion

From the perspective of the history of epigenetics, namely the long-recognised theory of separation of the somatic and germinal line. It is especially interesting to show TEI in mammals because in their case it has been rejected for a long time. Discussions about the relevance and even existence of TEI in mammals continue [126,127,128,129]. When considering TEI, there are two camps. There are those who believe it is a real phenomenon clearly in need of further research (e.g., authors cited in this review doing experiments on TEI) and, on the other hand, there are scholars who are sceptical of TEI, especially in case of humans [127] or considering studies related to vincozolin [130].

Scholars who do not believe in existence of TEI tend to emphasise genetic inheritance over the epigenetic one and often claim that epigenetic inheritance is almost always associated with the defence of sex cells against transposons, viruses, or transgenes. They often use the example of secondary epimutation, citing cases when DNA methylation of a gene is associated with mutation of a neighbouring gene [131] which removes the transcription termination signal. Consequently, expression of the mutated gene leads to abnormal promotor methylation and gene silencing, and that eventually leads to disease. When the mutated gene is active also in the germ cell, methylation can also take place but technically, this is not TEI. Some opponents of TEI claim that all epigenetic processing is gene-driven in a similar manner and that epigenetic marks do not survive reprogramming; therefore, they conclude one cannot speak about TEI [132]. Additionally, epimutations are believed to be rarely heritable or adaptive—they are usually considered to be merely noise [133].

On top of that, the pathways of signals from the brain to the sex cells are often unknown and the evidence regarding a causal mechanism of such transmission in often missing (e.g., the study on heritability of olfactory preferences [117]). Consequently, scientists are careful not to posit in their interpretation of results any causal links between signals from the environment, molecular data (such as methylation of certain sequences) and phenotypic traits. This applies especially to the impact of complex events, such as individual traumatic experience [80].

In defence of epigenetics, one should note that epigenetic modifications are reversible by definition, so we cannot expect the same stability as in genetic inheritance. Yet, such a stability requirement is inherently assumed when arguing that DNA methylation marks cannot be erased during cellular reprogramming if we are to speak about genuine epigenetic inheritance. In eukaryotic genetic inheritance, although the genetic transcript mutates, it is constantly stabilised by various processes in the cell and stably passed on to the next generation. The dynamics of epigenetic processes, however, is often difficult to capture. The modifications are numerous and some constantly rewritten and deleted. In short, some of these processes are inherently reversible, and therefore unstable. Often it is hard to decide whether they are induced randomly or in reaction to the environment. Moreover, incomplete epigenetic resetting is regarded as adaptive in fluctuating environments [134,135]. 

Nevertheless, the aforementioned exosomes loaded with RNA molecules—which, among other things, enrich sperm during their maturation in the epididymis—may be of crucial significance for future epigenetic research. In theory, if sex cells acquire regulatory RNA molecules from somatic cells, DNA in the sex cell itself may not carry as many memory traces (methylation and modification of histone proteins) because the RNAs themselves are capable of rewriting these memory traces in each successive generation. To make this even more complicated, Gapp and Bohacek [72] report that sperm RNAs found in fathers does not have to be the same in the offspring, although behavioural and metabolic changes were transmitted [121,136,137]. They hypothesize that sperm RNAs initiate TEI in the first transmission (F0 to F1), but in the transmission of the signal to the F2 generation another epigenetic process (e.g., chromatin modification) can take over.

## 7. Conclusions

Examples of transgenerational epigenetic inheritance show that changes in gene regulation, which can be inherited over several generations, respond plastically to environmental signals, and can therefore be thought of as acquired adaptations. We cannot speak of acquired traits, but rather of acquired states of gene regulation [138]. In addition, one should not extrapolate and conclude that all behaviour or experience are epigenetically inherited. Our knowledge of this topic is still focused mainly on experiences such as contact with environmental toxins, preparation for energy deficits, or other possible stress situations. It is yet to be seen to what extent this epigenetic variation has an impact on adaptive fitness or whether epivariation is just noise—as some claim. 

Jablonka [81] and others believe [6,139] that all these new findings are changing our view of evolution. They argue that epigenetic processes should be included in the theory of evolution as novel processes leading to the emergence of adaptive traits, as processes which may in fact represent a much faster and more plastic response to environmental changes than waiting for random mutations. Epigenetic processes can also explain the phenomenon of phenotypic accommodation by Eberhard [140], which unifies the Baldwin effect [141] and Waddington assimilation [142,143]. Epigenetic changes can also influence genetic changes. Methylated cytosines can affect the DNA sequence through spontaneous hydrolytic deamination of cytosine to uracil, which can—in a few additional steps and with the help of repair enzymes—lead to an A = T base pair in place of the previous G = C pair [144]. Hypermethylated genomic stretches then often correlate with higher mutation rates in these regions [145] and chromatin organization can contribute to mutation rates [146]. In addition, the abovementioned examples point to a missing link in the eternal nature vs. nurture debate, which may eventually become obsolete. TEI also has bearing on the discussion on possible evolution of instincts, where instinct is thought of as an originally learned behaviour that in a sense represents the memory of our ancestors [81,147,148].

To decide to what extent TEI is adaptive, especially in organisms with long generation times such as humans, will require more work in the field. However, there are already studies outside the focus of this review [149,150]. In addition, further studies on exosomes, which could help explain the communication between soma and germ, will be necessary. Such research should include genome- and epigenome-wide studies which could shed more light on the details of interactions among chromatin structure and its modifications, RNAs, and the genome.

## Figures and Tables

**Figure 1 genes-14-00120-f001:**
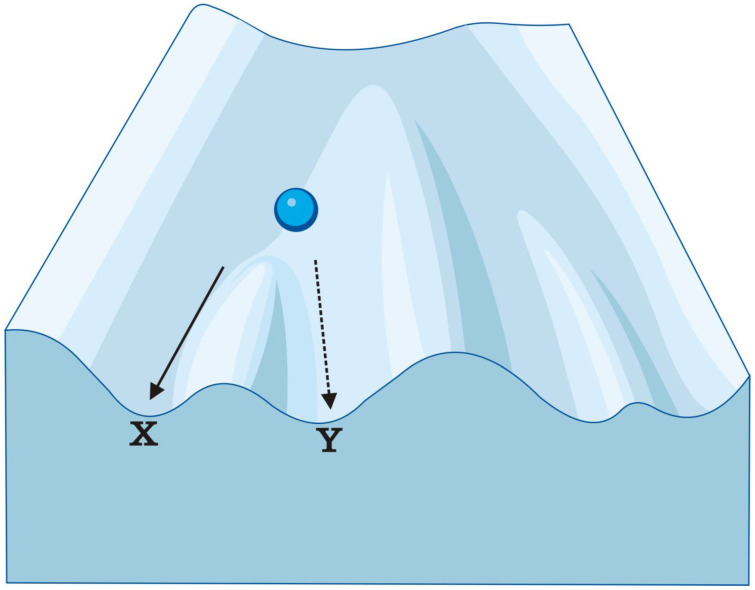
Epigenetic landscape by C.H.Waddington. Waddington’s metaphor of epigenetic landscape: the marble’s position in the landscape depicts the specific morphogenetic state which a cell/tissue/organ/organism can attain in possible morphospace during its development. It moves through an undulating landscape, which canalises the development into specific pathway (X) and makes it resistant to perturbations. The course of morphogenesis can change, i.e., an embryo can achieve to the same goal via more than one developmental pathway (Y).

**Figure 2 genes-14-00120-f002:**
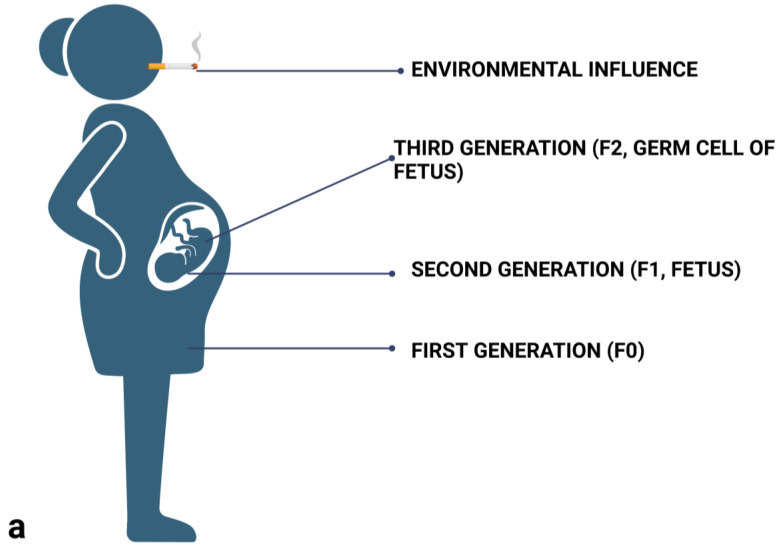
(**a**) In case of pregnant female, three generations can be simultaneously influenced by an external factor from the environment. Only in generation F3, however, can one consider the observed effect to be a genuine case of TEI. (**b**) In case of males, we can observe TEI already in the F2 generation.

**Figure 3 genes-14-00120-f003:**
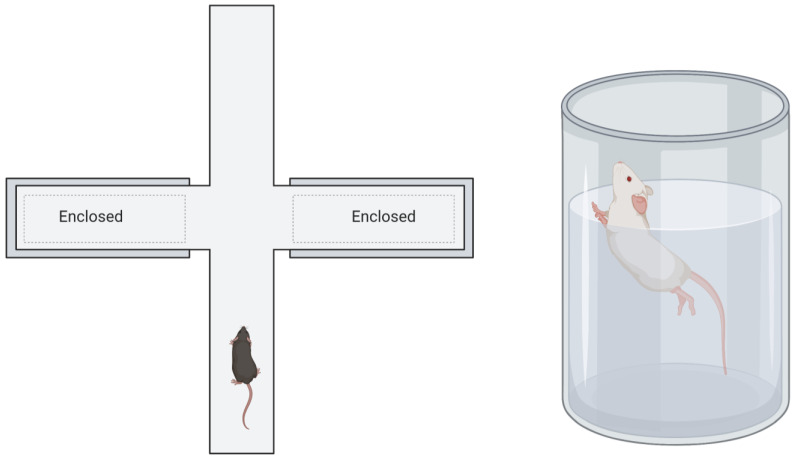
Behavioural testing in mice: the elevated plus maze (**left**) and the forced swim test (**right**). Elevated plus maze [82] is a construction that has two arms in the shape of the mathematical ‘+’ sign. One arm has walls, while the other has none and is thus in an open space. The test works on the assumption that rodents do not like open spaces and will spend more time in the closed arms when anxiety is increased, and vice versa. The forced swim test [83] is used to assess the level of depressed mood and potential effectiveness of antidepressants. A rodent is placed in a glass cylinder of moderately cold water for 15 min and what is measured is the amount of time the rodent spends swimming and attempting to free itself. Naturally, more depressed individuals spend less time swimming or attempting to escape. There are also behavioural tests which involve testing the recognition of novel objects or fear conditioning.

## Data Availability

Data sharing is not applicable to this article.

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
