# Peer review of "Transgenerational Epigenetic Inheritance of Traumatic Experience in Mammals"

_genes, 2023, doi:10.3390/genes14010120_

Round 1

Reviewer 1 Report

The present manuscript provides an excellent, comprehensive, and precise overview of transgenerational epigenetic modifications. It also emphatically discusses the differences between maternal and transgenerational epigenetic effects that are sometimes misunderstood. It would be of great interest to researchers interested in germline development, behavior, epidemiology, and many other fields, and I recommend publishing this review article. I would like to make a few comments.

Minor comments

  1. A more detailed explanation is needed for Figure 1. The labeling may be better incorporated into the illustration. 

  1. No explanation is given about genomic imprinting in the current manuscript (there is a brief description on page 4), which is a trans-generational epigenetic modification (DNA methylation and H3K27me3). In view of the fact that imprinting has been shown to affect both behavior and sociability and that it could be the cause of trans-generational epigenetic abnormalities, this reviewer thinks it would be helpful to provide a little more explanation. 

  1. P4 line 7: “But some regions, such as imprinted genes, intracisternal A-particle elements (IAPs), or LINE 1 elements, can apparently avoid such deletion [49].”

I find this sentence confusing. It is true that most imprinting marks are not demethylated during reprogramming at the preimplantation stage. A late-stage reprogramming in primitive germ cells by Tet1 removes the methylation of imprinting genes, allowing for the establishment of sperm- and oocyte-specific methylation patterns (PMID: 35431282, 24291790).

  1. P12 line 27-, or P14 Discussion: The study that conditioning trauma with acetophenone and electric shock transmits hypomethylation of the M71(Olfr151) gene to the child and affects its behavior is interesting, but the question of how odor stimulation and fear can reduce sperm methylation remains unclear. Most studies in this field face the same problem, which is not unique to this study. It is not yet clear how trauma and other stimuli affect the epigenetic state of the germline, despite the fact that the epigenetic state of the oocyte and sperm is already established by adulthood. It is important that the authors mention this unresolved critical question.

Author Response

I would like to thank to the reviewer for the very helpful and rightful comments. My response:

1) I have changed both the picture and its description.

2) In lines 141 to 152 there is an added text on genomic imprinting. 

3)  In lines 153 to 178 I also tried to explain better the two reprogramming events and explain in detail which loci remain methylated during these events. I cite studies which show the evidence for it.

4)It is true and I mentioned this type of criticism in the discussion. However, it was too vague, so I reformulated it with a direct reference to the study of Dias and Ressler. Please, see lines 530 to 535. Since it is a common problem, concerning more than one study, I would like to leave it in the discussion section.

I also found some minor mistakes and typos and added the model organisms in the descriptions of some of the studies. They were sometimes omitted and although it is mostly mouse model, it should be mentioned.

Also, I also moved two lines from 502 and 503 to 253 to 255

Thank you very much.

Best regards the author

Reviewer 2 Report

Transgenerational epigenetic inheritance is once controversial and is now supported by accumulating evidence. The author presents a great review of the history, biology, potential molecular mechanisms, and methodological challenges of related studies. This review is well-structured, thoughtful, informative and relevant.

I have two suggestions about the section on “DNA methylation”.               On page 3, in the statement: “5mC can be also oxidized by Tet proteins to 5-carboxy cytosine, 5-formyl cytosine, or 5-hydroxymethyl cytosine.”, the author may consider switching the order of 3 cytosine derivatives to “5-hydroxymethylcytosine, 5-formyl cytosine, and 5-carboxy cytosine” due to the sequential cascade of enzymatic oxidation by TETs proteins.

On page 4, in the sentence: “In mice, there are several transcription factors which bind histone modifications (H3K9me2), block methyl transferase Tet3 activity (which converts 5-methylcytosines to 5-hydroxymethylcytosines), and thus maintain DNA methylation marks.” “Methyl transferase” usually is used for DNA methyltransferases (DNMTs). The author may consider changing to “methylcytosine dioxygenase TET3”

Author Response

Thank you very much, I corrected the mistakes as advised.

Best regards the author